# Prevalence of metabolic syndrome and its association with cardiovascular disease risk and common risk factors amongst healthcare workers in Pakistan

Unab I. Khan[1☉‡*], Sonya F. Khan[2☉], Asra Qureshi[1‡]

1 Department of Family Medicine, Aga Khan University Hospital, Karachi, Sindh, Pakistan, 2 Department of Family Medicine, Ziauddin University Hospital, Karachi, Sindh, Pakistan

☉ These authors contributed equally to conceptualization, methodology, manuscript preparation and editing.
‡ UIK and AQ also contributed equally to Data curation, formal analysis and results.
* unab.khan@aku.edu

## Abstract

Metabolic syndrome (MetS) significantly increases the risk of cardiovascular disease (CVD). Healthcare workers (HCWs) are at a higher risk of CVD. However, little is known about the association between MetS and CVD risk in healthcare workers in Pakistan. We aimed to assess the prevalence of MetS and its components and examined the association between MetS and 10-year CVD-risk using Framingham Risk Score (FRS) and common CVD risk factors amongst HCWs working in a private healthcare system in Pakistan. This cross-sectional study uses baseline data from an existing CVD risk screening program for employees at a private healthcare system in Pakistan. MetS was diagnosed using the American Heart Association cut-offs for Asian population. Healthcare workers were divided into MetS positive and negative groups; demographics, MetS components and CVD risk were compared between these groups. Logistic regression was used to examine the association of MetS with 10-year CVD-risk and its risk factors. In 1,807 healthcare workers, 677 (37%) had MetS and 48 (2.7%) had a high 10-year CVD-risk. Of the MetS components, low High-density Lipoprotein (HDL) 1,467 (81%) and elevated waist circumference (WC) 1,049 (58%) were the most prevalent. Compared to MetS negative group, MetS positive group had a higher proportion of high-risk CVD (0.7% vs. 5.9%; *p*: <0.01). After controlling for known risk factors, we found that the odds of having MetS is 5.7 times higher (aOR: 5.67 (95% CI: 2.39–13.4) in those with high CVD risk. In addition, we found a significant association between screening positive for depression and MetS (OR: 2.42 (95% CI: 1.24–4.72). Interestingly, tobacco use was not significantly associated with MetS (OR: 0.81 (95% CI: 0.58–1.15). We found a high prevalence of MetS amongst Pakistani healthcare workers and of the MetS components, low HDL and elevated WC were the most prevalent. Along with biologic risk factors (age, sex and family history of CVD), depression significantly increases the odds of having MetS. In addition, both intermediate and high CVD risk groups have significant association with MetS. Comprehensive, workplace based screening and management programs are required for HCWs to mitigate the

**Data availability statement:** All relevant data are within the paper and its Supporting information files.

**Funding:** The study did not receive any grant or extramural funding. Internal funds provided by the Aga Khan University were used only to cover the costs of employees' assessments and laboratory investigations. None of the authors received any financial support. The University leadership had no role in study design, data collection and analysis, the decision to publish, or preparation of the manuscript.

**Competing interests:** The authors have declared that no competing interests exist.

risk of MetS and cardiovascular disease. Early identification and treatment of these risk factors may be cost-effective in lowering MetS burden in low-middle income countries.

## Introduction

Metabolic syndrome (MetS), a clustering of abdominal obesity, dyslipidemia, hyperglycemia and hypertension [1], is significantly associated with cardiovascular disease (CVD) [2]. In the Asia-Pacific region, Pakistan has the highest reported prevalence of MetS [3]. Despite having knowledge of health conditions, healthcare workers (HCWs), persons who directly or indirectly care for patients [4] have a higher prevalence of diabetes, hypertension and CVD compared to age-matched individuals from the general population [4–6]. Long working hours or shift work leads to physical inactivity during off-duty hours, unhealthy diet, smoking and mental stress [5]; are all factors that could potentially increase CVD risk in HCWs. Early identification and management of CVD risk factors could potentially decrease risk of CVD outcomes.

We assessed the prevalence of MetS and evaluated its association with the 10-year CVD-risk calculated using Framingham Risk Score (FRS) and other risk factors (such as age, gender, personal and family history of CVD, tobacco use and depression screening using Patient health questionnaire (PHQ2))) amongst HCWs in Pakistan.

## Materials and methods

### Study population

This is a sub-analysis of an ongoing 'Employee Health and Wellness Program' (EHWP), an employer-sponsored, health-screening program at Aga Khan University, Pakistan (AKU). AKU is a private, not-for-profit institution with 12,300 employees in Pakistan. EHWP offers free-of-cost screening and health assessment for early identification and timely referral for common CVD risk factors and depression. Details of EHWP can be found elsewhere [7]. The study was approved by the institutional Ethics Review Committee (2019-1281-3520). All HCWs signed informed consent.

For this sub-analysis, we used baseline data of HCWs who participated in EHWP from May 2019 to March 2020. Of the 5,286 HCW invited for free health screening, 1,807 completed health risk assessment.

### Definitions

We used the following definitions for assessment and diagnosis.

### Metabolic syndrome (MetS)

We used the American Heart Association/National Heart, Lung, and Blood Institute (AHA/NHLBI) cut-off values for Asian population to define MetS [8]. A HCW was considered to have MetS if they had three or more of the following five components: (i) Elevated waist circumference (men ≥90 cm, women ≥80 cm); (ii) Elevated Triglycerides: ≥150 mg/dl; (iii) Reduced HDL: < 40 mg/dl in men, < 50 mg/dl in women or on treatment for dyslipidemia; (iv) Elevated fasting blood glucose: ≥100 mg/dl or history of pre-existing diabetes; (v) Elevated blood pressure (BP ≥130 and/or ≥85 mmHg) or taking antihypertensive medications.

### Framingham risk score (FRS)

The Framingham risk score (FRS) offers a gender-specific prediction tool for the assessment of 10-year CVD-risk and has been validated for CVD risk assessment in Indian adults [9]. We

used the Framingham 2008 general cardiovascular disease equation to assess the 10-year CVD risk [10]. This score is based on six coronary risk factors including age, gender, total cholesterol, HDL cholesterol, systolic and diastolic blood pressure, smoking and the presence of diabetes. The HCWs were then classified as low-risk (< 10%), intermediate-risk (10– < 20%), and high-risk (≥ 20%) for developing CVD over the next 10 years [11].

## Blood tests

Blood samples were collected after 12 hours of fasting. Plasma glucose level was measured using the hexokinase method. Lipid profile was determined using standard enzymatic procedures. Blood samples were analyzed at the Aga Khan University Hospital laboratory, which is certified by the College of American Pathologist (CAP).

## Demographic and medical information

Age, gender, personal history of hypertension, diabetes, cardiovascular disease (heart attack and stroke) were obtained directly from the HCWs. Family history was defined as positive if the HCW reported history of ischemic heart disease, stroke, hypertension, or diabetes in first-degree relatives. Expecting resistance from the community in answering mental health questions, we initially used Wholey's questionnaire for depression screening [12,13]. For this manuscript, we have defined a positive screen for depression in participants who answered "Yes" to either of the two questions on Wholey's questionnaire or had a score of 3 or greater on PHQ-2 [14]. The sensitivity of Wholey's questionnaire and PHQ-2 is similar in screening for depression [12]. HCW who responded positively to presently smoking cigarettes, e-cigarettes, hookah and/or use of smokeless tobacco such as chewing tobacco were considered as having positive tobacco use.

## Anthropometric measures

Height in centimeters and weight in kilogram were measured in light clothing and without shoes. Body mass index (BMI) was calculated as weight in kilograms divided by height in meters squared. Waist circumference was measured in centimeters in non-restrictive clothing, using a non-stretchable fiber measuring tape, midway between the costal margin and the iliac crest at minimal respiration. Blood pressure was measured in the right arm with the HCW seated, after at least 5 minutes of rest using an automatic oscillometer blood pressure monitor.

## Statistical analysis

Descriptive statistics are reported as mean ± SD for normally distributed variables and median ± IQR for variables that were not normally distributed. For categorical data, frequency and proportions are reported.

To examine differences between MetS positive and MetS negative groups, we used Chi-square for categorical variables and t-test for normally distributed continuous variables. For skewed data, we used Mann-Whitney U test.

The prevalence of MetS components across CVD risk categories was examined to determine statistical difference.

Finally, to examine the association between MetS and 10-year CVD risk using FRS, we created logistic regression models, adding biologic factors, lifestyle risk factors and depression in a stepwise fashion. Adjusted Odds Ratio (aOR) with 95% CI was reported. All analyses were done using Stata/SE 15.1.

## Results

Baseline characteristics comparing MetS positive and MetS negative HCWs are presented in Table 1. Of the 1,807 HCWs, 677 (37.4%) had MetS. Compared to HCWs in MetS negative group, the MetS positive group was older (40 ± 9.2 vs. 43.5 ± 9.3; $p$: < 0.01); and had a higher percentage of male participants (73% vs. 86.4%; $p$: < 0.01). Of the MetS components, low

**Table 1. Baseline characteristics of healthcare workers.**

| Characteristics | Total N 1,807 | MetS positive n (%) 677 (37.4%) | MetS negative n (%) 1,130 (62.5%) | p-value |
|---|---|---|---|---|
| **Age** (years) mean ±SD | 39.4 ± 9.8 | 43.5 ± 9.3 | 40 ± 9.2 | <0.01* |
| **Sex** % Male | 1,411 (78) | 585 (86.4) | 826 (73) | <0.01† |
| **Systolic blood pressure** (mean ± SD) mmHg | 122.6 ± 12.4 | 130 ± 12.0 | 118 ± 10.4 | <0.01* |
| **Diastolic blood pressure** (mean ± SD) mmHg | 76.7 ± 8.7 | 81.3 ± 8.2 | 74 ± 7.9 | <0.01* |
| **Elevated blood pressure** (≥130/85mmHg) or on anti-hypertensives | 589 (32.6) | 441 (65) | 148 (13) | <0.01† |
| **Waist circumference** (median ± IQR) cm | 90 (84-96) | 96 (92-100) | 86 (80-92) | <0.01‡ |
| **Elevated WC** (female ≥80 cm; male ≥90 cm) | 1,049 (58) | 606 (89.5) | 443 (39.2) | <0.01† |
| **Fasting glucose** (median ± IQR) mg/dl | 87 (82-94) | 91 (85-105) | 85 (81-90) | <0.01‡ |
| **Elevated fasting glucose** (≥100 mg/dl) or on anti-diabetics | 263 (13.7) | 228 (33.6) | 35 (3.1) | <0.01† |
| **Triglycerides** (median ± IQR) mg/dl | 118 (84-169) | 171 (130-231) | 97 (74-127) | <0.01‡ |
| **Elevated triglycerides** (≥150 mg/dl) | 594 (33) | 454 (67) | 140 (12.3) | <0.01† |
| **HDL** (median ± IQR) mg/dl | 35 (31-41) | 33 (30-37) | 37 (32-43) | <0.01‡ |
| **Low HDL** Female (≤50 mg/dl) Male (≤40 mg/dl) | 1,467 (81) | 643 (95) | 824 (73) | <0.01† |
| **Obesity** (overweight/obese) BMI≥25 Kg/m2 | 1,026 (56.7) | 560 (82.7) | 466 (41.2) | <0.01† |
| **Tobacco use** (cigarettes/ smokeless) | 249 (13.7) | 107 (16) | 142 (12.5) | 0.053† |
| **PHQ-2** | 52 (2.8) | 28 (4.1) | 24 (2.1) | 0.01† |
| **Family history for CVD** | 336 (18.5) | 152 (22.4) | 184 (16.2) | 0.01† |
| **10-year CVD risk using Framingham Risk Score** Low Risk | 1,526 (84.5) | 472 (70.0) | 1,054 (93.3) | |
| Intermediate Risk | 233 (13.0) | 165 (24.3) | 66 (6.0) | <0.01§ |
| High Risk | 48 (2.7) | 40 (5.9) | 8 (0.7) | |

MetS = Metabolic Syndrome; HDL = High Density Lipoprotein; BMI = Body Mass Index; PHQ-2 = Patient Health Questionnaire-2 (Wholey Questionnaire to screen Depression); CVD = cardiovascular disease.

*t-test for two independent samples.

†$\chi^2$ test of independence.

‡Mann-Whitney U test.

§Kruskal Wallis equality of population rank test.

HDL 1,467 (81%) and elevated waist circumference 1,049 (58%) were the most prevalent. As expected, MetS positive group had higher systolic and diastolic blood pressures, waist circumference, fasting blood glucose levels and triglyceride levels, and lower serum HDL levels. In addition, as compared to MetS negative group, a higher percentage of MetS positive group had intermediate and high 10-year CVD-risk according to FRS (6% vs. 24.3%; *p*: 0.01) and (0.7% vs. 5.9%; *p*: 0.01) respectively.

We examined the association between each MetS component across CVD-risk categories Table 2. We found that prevalence of elevated blood pressure, elevated fasting blood glucose and elevated triglycerides increased with increasing 10-year CVD risk. However, the prevalence of elevated waist circumference and low HDL was highest in intermediate CVD risk category.

Table 3 shows the multivariate association of MetS with 10-year CVD-risk using FRS and other risk factors. We see that compared to women, male HCWs are at almost three times higher odds (aOR: 2.82 (95% CI: 2.05–3.89) of having MetS. Similarly, there is a 1.24 times increased odds (aOR: 1.24 (95% CI: 1.21–1.28) of being MetS positive with each 1 kg/m2 unit increase in BMI. A positive family history of CVD was associated with a higher odds of having MetS (aOR: 1.44 (95%CI: 1.08-1.92)). After controlling for CVD risk factors, we find that the odds of having MetS are 2.5 times higher (aOR: 2.52 (95% CI: 1.73–3.68) in HCWs with intermediate CVD risk and 5.7 times higher (aOR: 5.68 (95% CI: 2.40–13.45) in those in high 10-year CVD risk category. Interestingly, there is a significant association between positive depression screening and presence of MetS (OR: 2.42 (95% CI: 1.24–4.72).

## Discussion

Metabolic Syndrome increases the risk of cardiovascular disease and has been studied extensively in general population. With high CVD prevalence in healthcare workers (HCWs), it is important to identify all risk factors including the presence of MetS. This study is one of the very few studies investigating the prevalence of MetS amongst HCWs in Pakistan as well as evaluating the association of MetS with 10-year risk of developing CVD.

We found a higher prevalence of MetS in our cohort of healthcare workers (37%) compared to the general population (28.8%) in Pakistan [15] showing the importance of early identification and treatment of this group. MetS positive group had a higher proportion of high 10-year CVD-risk as compared to MetS negative group (5.9% vs. 0.7%; *p* - < 0.05).

**Table 2. Prevalence of Metabolic Syndrome components in 10-year CVD risk categories* in healthcare workers.**

| Components | Total N = 1,807 | Low Risk 1,526 (84%) | Intermediate Risk 233 (13%) | High Risk 48 (3%) | *p*-value** |
|---|---|---|---|---|---|
| **Elevated Blood Pressure** (≥130/85mmHg) or on antihypertensives | 589 (32) | 394 (26) | 158 (68) | 37 (77) | <0.001 |
| **Elevated Waist Circumference** (female ≥80 cm, male ≥90 cm) | 1,049 (58) | 851 (55) | 165 (71) | 33 (69) | <0.001 |
| **Elevated Fasting Glucose** (≥100 mg/dl) or on antidiabetic medications | 263 (14) | 143 (9.3) | 86 (37) | 34 (71) | <0.001 |
| **Elevated Triglycerides** (≥150 mg/dl) | 594 (33) | 456 (30) | 112 (48) | 26 (54) | <0.001 |
| **Low HDL** (Female ≤50 mg/dl, Male ≤40 mg/dl) | 1,467 (81) | 1,218 (80) | 209 (89) | 40 (83) | 0.001 |

HDL = High-Density Lipoprotein.

*CVD risk calculated using Framingham Risk Score 2008 generalized equation.

**Chi-square test of independence.

**Table 3. Multivariate Logistic Regression Analysis showing associations between metabolic syndrome, CVD risk factors and 10-year CVD risk using Framingham risk score (FRS).**

| Predictors | Model 3 N = 1,807 LR Chi2 = 545.36 Pseudo R2 = 0.22 | | |
| --- | --- | --- | --- |
| | aOR | 95%CI | p-value |
| **Sex** | | | |
| Female | – | – | – |
| Male | 2.82 | 2.05–3.89 | 0.00 |
| **Age (years)** | 1.03 | 1.01–1.04 | 0.00 |
| **10-year CVD Risk*** | | | |
| Low Risk (0-< 9.99%) | – | – | – |
| Intermediate Risk (10- < 19.99%) | 2.52 | 1.72–3.67 | 0.00 |
| High Risk (≥ 20%) | 5.67 | 2.39–13.4 | 0.00 |
| **Tobacco Use** | | | |
| No | – | – | – |
| Yes | 0.81 | 0.58–1.15 | 0.24 |
| **BMI (kg/m²)** | 1.24 | 1.21–1.28 | 0.00 |
| **Family History of CVD**** | | | |
| No | – | – | – |
| Yes | 1.44 | 1.08–1.92 | 0.011 |
| **PHQ – 2*****  | | | |
| No | – | – | – |
| Yes | 2.42 | 1.24–4.72 | 0.009 |

p - value < 0.05 is considered as significant.

*10-year CVD risk using Framingham Risk Score 2008 generalized equation.

**Family History of Cardiovascular Disease in first- degree relative.

***Patient Health Questionnaire – 2 (Wholey Questionnaire for depression screening).

The most frequently seen MetS components in HCWs was low HDL levels (81%) and central obesity (58%); followed by elevated TGs (33%). A study conducted in the general population of Pakistan reported a prevalence of low HDL at 48%, central obesity at 37%, and elevated TGs at 36% [15]. This means that HCWs are at an increased risk of metabolic abnormalities that can significantly increase CVD risk as compared to the general population.

While smoking is a known modifiable risk factor for CVD and MetS [16,17], we did not find a significant difference in the prevalence of smoking among MetS positive and negative HCWs. In addition, in multivariate analysis, tobacco use was not associated with MetS. Similar results were reported in a population study in Iran [18]. The diminished appetite, desensitization of taste buds and augmented metabolism in tobacco users [18,19] may be related to this unexpected finding.

Depression is a known risk factor of MetS and increased CVD risk [20,21]. Published literature establishes that HCWs are susceptible to high levels of psychological distress [22] and anxiety [23]. Prevalence of depression amongst HCWs ranges from 21.53% to 32.77%, much higher than that of the general population worldwide (4.40% in 2015) [24–27]. While a very small percentage of HCWs screened positive for depression in our study, we did find that there was a higher prevalence of depression in MetS positive as compared to MetS negative group (p – 0.01). More importantly, we found that after adjusting for other CVD risk factors,

screening positive for depression increased the odds of having MetS two folds (OR: 2.42 (95% CI: 1.24–2.72). Thus, mental health concerns need to be addressed in HCWs as part of comprehensive CVD risk reduction.

While our study provides valuable insight, it is important to acknowledge its limitation. Our study has limitations. The cross-sectional design precludes any comment on the direction of association between MetS and CVD-risk. While HCWs were from across the country, they were all part of a single, private healthcare system, which may preclude generalizability of results to HCWs across Pakistan.

Despite these limitations, our study provides useful insights in CVD risk among healthcare workers in Pakistan. The high prevalence of MetS (37%) in a relatively young cohort of healthcare workers and elevated 10-year CVD risk (15%) are a cause for concern. This shows that HCWs are at a high risk of adverse CVD events. Our study builds a case for comprehensive CVD risk assessment and management in workplace-based programs to care for this important cohort of people who take care of others.

## Conclusion

MetS increases the risk of cardiovascular disease. Our study provides evidence that HCWs have a high prevalence of MetS and are at an increased risk of CVD. In addition, screening and treating depression in HCWs may be a cost-effective way of decreasing CVD risk in HCWs.

## Acknowledgments

The authors would like to acknowledge the continuing support of the following entities at the organization: Human Resources Department; Information Technology and Dept. of Software Development and Maintenance; and Employee Health team at Aga Khan University.

## Author contributions

**Conceptualization:** Unab I. Khan.

**Data curation:** Unab I. Khan, Asra Qureshi.

**Formal analysis:** Unab I. Khan, Sonya F. Khan, Asra Qureshi.

**Funding acquisition:** Unab I. Khan.

**Investigation:** Unab I. Khan.

**Methodology:** Unab I. Khan, Sonya F. Khan.

**Writing – original draft:** Sonya F. Khan.

**Writing – review & editing:** Unab I. Khan, Sonya F. Khan.

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
