## [Decision Letter · Decision Letter 0]

18 Oct 2024

PGPH-D-24-01994

Prevalence of metabolic syndrome and its association with cardiovascular disease risk and common risk factors amongst healthcare workers in Pakistan

Dear Dr. Khan,

Thank you for submitting your manuscript to PLOS Global Public Health. After careful consideration, we feel that it has merit but does not fully meet PLOS Global Public Health’s publication criteria as it currently stands. Therefore, we invite you to submit a revised version of the manuscript that addresses the points raised during the review process.

As outlined by Reviewer-1 both FRS and MetS are indicators of CVD risk. As such, the direction of this study and its contribution to the literature remain ambiguous. Please carefully address this concern (along with other reviewer comments) and revise the manuscript accordingly including the study aims if required.

We look forward to receiving your revised manuscript.

Kind regards,

Biplab Datta, Ph.D.

Academic Editor

Journal Requirements:

Additional Editor Comments (if provided):

Reviewers' comments:

Reviewer's Responses to Questions

**Comments to the Author**

1. Does this manuscript meet PLOS Global Public Health’s publication criteria ? Is the manuscript technically sound, and do the data support the conclusions? The manuscript must describe methodologically and ethically rigorous research with conclusions that are appropriately drawn based on the data presented.

Reviewer #1: Partly

Reviewer #2: Yes

2. Has the statistical analysis been performed appropriately and rigorously?

Reviewer #1: I don't know

Reviewer #2: Yes

3. Have the authors made all data underlying the findings in their manuscript fully available (please refer to the Data Availability Statement at the start of the manuscript PDF file)?

Reviewer #1: Yes

Reviewer #2: Yes

4. Is the manuscript presented in an intelligible fashion and written in standard English?

Reviewer #1: Yes

Reviewer #2: Yes

5. Review Comments to the Author

Reviewer #1: Thank you for the opportunity to review this manuscript. The aim of the study, which is to analyze the cardiovascular disease (CVD) risk among healthcare workers (HCWs), is a valuable addition to the literature. Below are my major and minor comments on the manuscript:

Major Comment: The study provides important insights by analyzing the CVD risks among HCWs and comparing them with overall population outcomes from the literature. However, it is unclear why the authors analyzed the association between the Framingham Risk Score (FRS) and Metabolic Syndrome (MetS) among HCWs. Literature (Wannamethee SG, Shaper AG, Lennon L, Morris RW. Metabolic syndrome vs Framingham Risk Score for prediction of coronary heart disease, stroke, and type 2 diabetes mellitus. Archives of Internal Medicine. 2005 Dec 12;165(22):2644-50.) indicates that both FRS and MetS are calculators of CVD risks. The authors need to discuss their hypothesis about how one determines the other and describe the objective behind using these two indicators as the dependent and independent variables in the logistic regression model.

Minor Comment: One of the limitations mentioned in the manuscript pertains to the generalizability of the results. The statement in lines 40-41 of the abstract should be edited to better reflect this limitation.

Reviewer #2: 1. The conclusion in the abstract is too lengthy. 

2. Multivariate logistic regression table is showing only model 3. why not model 1 and 2.

3. Add updated literature in references. There is no study from 2024.

4. The study has reported the MetS is higher (37%) in HCWs compared to the general population (28%). However, the reason for this higher frequency of HCW is not reported. The study could be improved by identifying/reporting the possible reason for this higher percentage in HCWs compared to the general population.  Secondly, HCWs include doctors, nurses and paramedics. Which particular HCWs are more at risk should be highlighted. if this information is not available, it should be highlighted in the limitation of the discussion section.

5. A considerably higher elevated waist circumference, overweight and obesity along with elevated low HDL are very strong findings. Particularly again in the HCWs this is a public health concern and should be reported. 

6. The study states that HCWs from all over Pakistan are included in the data set. However, there is no information available about the pattern of the disease distribution with respect to province or city as well.

6. PLOS authors have the option to publish the peer review history of their article (what does this mean? ). If published, this will include your full peer review and any attached files.

**Do you want your identity to be public for this peer review?** For information about this choice, including consent withdrawal, please see our Privacy Policy .

Reviewer #1: No

Reviewer #2: **Yes: ** Syed Omair Adil

---

## [Decision Letter · Decision Letter 1]

19 Dec 2024

Prevalence of metabolic syndrome and its association with cardiovascular disease risk and common risk factors amongst healthcare workers in Pakistan

PGPH-D-24-01994R1

Dear Dr. Khan,

We are pleased to inform you that your manuscript 'Prevalence of metabolic syndrome and its association with cardiovascular disease risk and common risk factors amongst healthcare workers in Pakistan' has been provisionally accepted for publication in PLOS Global Public Health.

Best regards,

Biplab Datta, Ph.D.

Academic Editor

Reviewer Comments (if any, and for reference):

Reviewer's Responses to Questions

**Comments to the Author**

1. If the authors have adequately addressed your comments raised in a previous round of review and you feel that this manuscript is now acceptable for publication, you may indicate that here to bypass the “Comments to the Author” section, enter your conflict of interest statement in the “Confidential to Editor” section, and submit your "Accept" recommendation.

Reviewer #1: All comments have been addressed

Reviewer #2: All comments have been addressed

2. Does this manuscript meet PLOS Global Public Health’s publication criteria ? Is the manuscript technically sound, and do the data support the conclusions? The manuscript must describe methodologically and ethically rigorous research with conclusions that are appropriately drawn based on the data presented.

Reviewer #1: Yes

Reviewer #2: Yes

3. Has the statistical analysis been performed appropriately and rigorously?

Reviewer #1: Yes

Reviewer #2: Yes

4. Have the authors made all data underlying the findings in their manuscript fully available (please refer to the Data Availability Statement at the start of the manuscript PDF file)?

Reviewer #1: Yes

Reviewer #2: Yes

5. Is the manuscript presented in an intelligible fashion and written in standard English?

Reviewer #1: Yes

Reviewer #2: Yes

6. Review Comments to the Author

Reviewer #1: Thank you for addressing the comments made.

Reviewer #2: All my previous comments are incorporated. There are no further comments.

7. PLOS authors have the option to publish the peer review history of their article (what does this mean? ). If published, this will include your full peer review and any attached files.

**Do you want your identity to be public for this peer review?** For information about this choice, including consent withdrawal, please see our Privacy Policy .

Reviewer #1: No

Reviewer #2: **Yes: ** Syed Omair Adil
